behaviour/cognition/neuroscience

social behaviour, cognition, therapy adherence, prisoners' dilemma game, functional magnetic resonance imaging

**Author for correspondence:**
Andrija Javor
e-mail: javor@gmx.at

†These authors contributed equally to this study.

### PUBLISHING

# Social cognition, behaviour and therapy adherence in frontal lobe epilepsy: a study combining neuroeconomic and neuropsychological methods

Andrija Javor[1,†], Carolina Ciumas[2,3,5,†], Danielle Ibarrola[4], Philippe Ryvlin[3,5,†] and Sylvain Rheims[2,5,6,†]

[1]Department of Neurology 2, Kepler University Clinic, Linz, Austria
[2]Translational and Integrative Group in Epilepsy Research (TIGER), INSERM U1028, CNRS UMR5292, Lyon Neuroscience Research Center, University Lyon 1, Lyon, France
[3]Department of Clinical Neurosciences, CHUV, Lausanne, Switzerland
[4]CERMEP - Imagerie du vivant, MRI department and CNRS UMS3453, Lyon, France
[5]Epilepsy Institute (IDEE), Lyon, France
[6]Department of Functional Neurology and Epileptology, Hospices Civils de Lyon and University of Lyon, Lyon, France

AJ, 0000-0002-6285-8104

Social behaviour of healthy humans and its neural correlates have been extensively studied in social neuroscience and neuroeconomics. Whereas it is well established that several types of epilepsies, such as frontal lobe epilepsy, lead to social cognitive impairments, experimental evidence on how these translate into behavioural symptoms is scarce. Furthermore, it is unclear whether social cognitive or behavioural disturbances have an impact on therapy adherence, which is critical for effective disease management, but generally low in these patients. In order to investigate the relationship between social cognition, social behaviour, and therapy adherence in patients with frontal lobe epilepsies (FLE), we designed a study combining conventional neuropsychological with behavioural economic and functional magnetic resonance imaging (fMRI) methodology. Fifteen patients and 15 healthy controls played a prisoners' dilemma game (an established game to operationalize social behaviour) while undergoing fMRI. Additionally, social cognitive, basic neuropsychological variables, and therapy adherence were assessed. Our results implicate that social behaviour is indeed affected and can be quantified using neuroeconomic methods in patients with FLE.

Impaired social behaviour in these patients might be a consequence of altered brain activation in the medial prefrontal cortex and play a role in low therapy adherence. Finally, this study serves as an example of how to integrate neuroeconomic methods in neurology.

## 1. Introduction

Social cognition is a term used for several high-level cognitive functions that determine human behaviour in a social context. Several studies showed impairments of social cognition in frontal lobe epilepsies (FLE), such as Theory-of-Mind/mentalizing and facial emotion recognition [1], as well as neuropsychiatric co-morbidities [2], but studies quantifying social behaviour abnormalities of FLE patients in an experimental sense are scarce. Even though behaviour relies on cognitive functions, the relationship between cognition, behaviour and epilepsy variables is complex and not fully understood [3], making it difficult to predict the impact of altered social cognition on social behaviour, and more specifically, therapy adherence in FLE.

In neuroeconomics—the science of studying research questions in economics through the application of neuroscientific methods and theories—pro-social behaviours, such as trust or cooperation, have been extensively studied for decades, mainly through game paradigms, such as the trust game or the prisoners' dilemma game [4]. Using this approach, several brain areas were found to play a role in pro-social behaviour of healthy humans. Crucial parts of this brain network reside in the frontal cortex such as the ventromedial frontal/orbitofrontal cortex and the anterior cingulate cortex [5]. These discoveries have led to the adoption of economic methodology to study behaviour in neurology [6], but to the best of our knowledge, no such study has focused on the behaviour of patients with epilepsy.

Adherence to anti-epileptic drug therapy is critical for effective disease management. Although its measurement is difficult without a single method that has yet proved to be the gold standard [7], therapy adherence of patients suffering from epilepsy is low (at about 30–50%) [8]. This is unfortunate, as low therapy adherence not only leads to poorer seizure control, but also increases the risk of sudden unexpected death in epilepsy [9]. The reasons for low therapy adherence in general are still a matter of research [10]. Low therapy adherence seems to have a multifactorial origin with some factors being associated with neuropsychological impairments and psychiatric co-morbidities of chronic diseases [11], and others reflecting pathophysiological changes of neural networks specifically affected in epilepsy [12].

We thus designed a study combining conventional neuropsychological with neuroeconomic methods to address the following open research questions:

(i) Is there a difference in social behaviour between FLE patients and healthy controls that can be experimentally operationalized?
(ii) If such a difference exists, is it associated with altered social cognitive functions and their underlying frontal lobe network?
(iii) Does social behaviour affect therapy adherence in FLE?

## 2. Methods

### 2.1. Subjects

We included 15 FLE patients and 15 healthy controls. We established the following inclusion and exclusion criteria to ensure the feasibility of the study and avoid known confounders in behavioural research.

Inclusion criteria for patients were: age between 18 and 50, right-handedness (Edinburgh handedness inventory; [13]), no significant anxiety, depression, or obsessive-compulsive symptoms as assessed by a hospital anxiety and depression scale score (HADS) less than 10 [14] and an obsessive compulsive inventory score (OCIS) less than 40 [15], sufficient language skills, and a diagnosis of FLE. The latter was based either upon association between typical frontal lobe seizure semiology and existence of an epileptogenic MRI lesion within the frontal lobe or upon direct recording of seizures of frontal lobe origin during a long-term video-EEG monitoring performed in the Department of Functional Neurology and Epileptology at Hospices Civils de Lyon, France, in patients with normal MRI. Exclusion criteria included pregnancy, non-MRI suitable transplants, major perceptive impairments, non-epileptic seizures, a history of intellectual disability, other known neurological diseases, MRI-

**Table 1.** Pay-off structure of the prisoners' dilemma game.

|  |  | COUNTERPART | |
|---|---|---|---|
|  |  | cooperate | defect |
| PARTICIPANT | cooperate | 30<br><br>30 | 50<br><br>0 |
|  | defect | 0<br><br>50 | 10<br><br>10 |

lesion outside of the frontal lobe or frontal cortical lesions larger than 1 cm in diameter, and past epilepsy or other brain surgery in order to achieve a patient sample of non-resected participants.

Inclusion criteria for controls were: age between 18 and 50, right-handedness (Edinburgh handedness inventory; [13]), no significant anxiety, depression, or obsessive-compulsive symptoms assessed by a HADS Score less than 10 [14] and an OCIS less than 40 [15] as well as sufficient language skills. Exclusion criteria were pregnancy, non-MRI suitable transplants, major perceptive impairments, medication other than contraceptives, or a history of neurological or psychiatric diseases.

## 2.2. Magnetic resonance data acquisition

All images were acquired using the same MRI machine (Siemens Magneton Prisma 3 Tesla) in one session per participant. Structural MRI and shimming (to minimize field inhomogeneities) were performed on all subjects prior to gradient-echo echoplanar imaging that provided blood oxygen level dependent (BOLD) contrast. Each volume comprised 35 AC-PC aligned slices (order of acquisition: interleaved) with a thickness of 2.5 cm, field of view (FOV) 23 cm, parallel imaging parameters GRAPPA/acceleration factor 2, echo time (TE) 26 ms, repetition time (TR) 2260 ms.

## 2.3. Paradigm

We designed our paradigm on the basis of similar experiments in the literature [5] to ensure comparability. When subjects entered the laboratory for the experiment, they received written instructions explaining the prisoners' dilemma game (PDG) [4]. In this game, two players decide at the same time whether or not to cooperate with the other player. Depending on the decisions of both players, there is a monetary pay-off, which is equal for both players in the case of mutual cooperation or defection, whereas in the case of divergent strategies, it is not existent for the cooperating player and highest for the defecting player. The pay-off structure of our version of the game is shown in table 1.

Participants were told that they would play with real money and be paid out their gain. Payment took place at the end of the experiment, but in fact, all participants received the same amount of money (40€) independent of their actual gain due to ethical considerations. The participants did not know the number of rounds played in the game, but in fact all subjects played 32 rounds. Moreover, participants (player 1) were told that they would be playing with four different human beings (two males and two females—here termed player 2) located in another room, but actually played against a computer-generated, randomized, strategy that simulated the other player's behaviour. This means that each participant played 32 rounds with each of the four different counterparts (equals 128 decisions to cooperate or to defect in total). Player 2 was illustrated with face images from an established database (Glasgow Unfamiliar Face Database). Once installed in the MRI machine, participants first had to rate the trustworthiness of the presented player 2 on a Likert-scale between 1 and 7 (1 no trustworthiness, 7 highest possible trustworthiness) being shown a face picture of player 2. Next, the actual game began. Each round of the game consisted of 3 screens: first, the pay-off matrix of the game was shown for 2 s. Then, participants were asked to choose to either cooperate or to defect using two input buttons. After a random time interval between 4 and 6 s simulating variable decision times of player 2, the result of the round was shown for 2 s using the pay-off matrix with the result highlighted in yellow colour. After 32 rounds of the game, the face of player 2 was again shown to the participants, who had to rate the trustworthiness of the

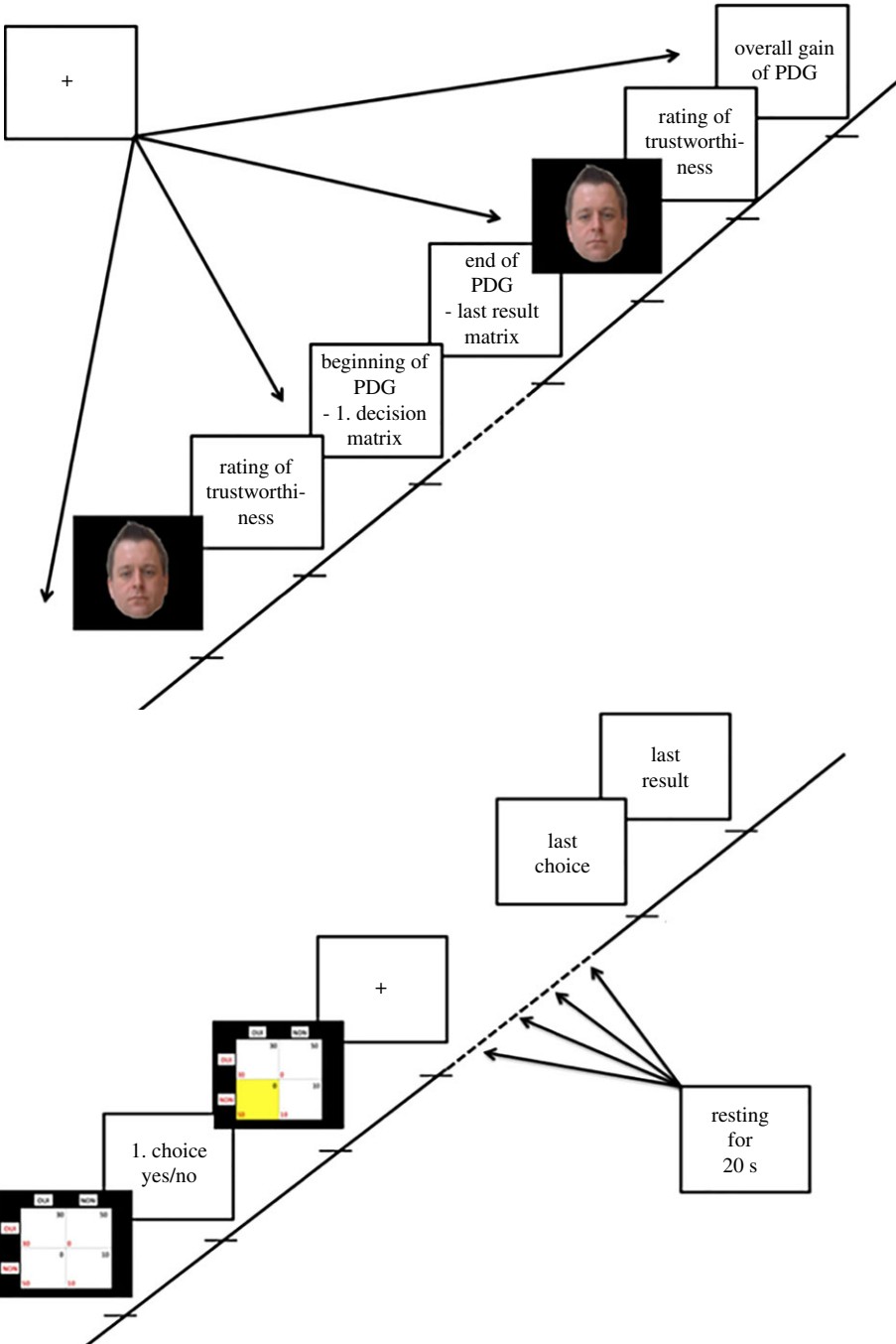

**Figure 1.** Visualization of the experimental paradigm. Notes: top left—visualization of the beginning and end of the paradigm, bottom right—visualization of the time period in between (corresponds to dashed line in the top left part of the figure); PDG, prisoners' dilemma game.

face another time in analogy to the start. Finally, the overall gain of the game was shown. See figure 1 for a visualization.

## 2.4. Demographic data, neuropsychological testing, questionnaires and pill counts

We recorded the following variables in all participants via a self-report questionnaire: age, gender, profession and education. For patients, we included the number of seizures (generalized tonic-clonic seizures and other seizure types separately) in the last three months according to a seizure calendar commonly used in clinical routine and current anticonvulsive treatment (number and names of drugs, number of intakes per day, preparation of drugs for intake by the participant or by a carer) as recorded in the patient's medical chart.

Neuropsychological variables captured in all participants included psychomotor speed and mental flexibility (Trail Making test A and B [16]), memory for faces and working memory (Faces subsets and numbers of the Wechsler Memory Scale, fourth edition [17]).

Social cognition was tested through the Reading the Mind in the Eyes test [18] for Theory-of-Mind/mentalizing (in this test, subjects have to choose the correct word out of a list describing emotions to corresponding photos of a person's expression of eyes) and a trust questionnaire ([19]; French version), additionally to the trustworthiness ratings of faces during the fMRI paradigm.

Furthermore, therapy adherence was measured in the patient group through pill counts at two consecutive visits [20] by reviewing pill bottles of a six months period and several questionnaires comprising the Morisky adherence scale ([21,22]; validated French version), the Beliefs about Medicines Questionnaire (BMQ; [23,24]; validated French version) and the SATMED-Q (Treatment Satisfaction with Medicines Questionnaire, [25]; French version). These questionnaires are commonly used instruments in research on therapy adherence.

## 2.5. Data analysis

### 2.5.1. Imaging data

fMRI data were analysed using SPM 12 (https://www.fil.ion.ucl.ac.uk/spm/). Pre-processing involved slice time correction, realignment, normalization and smoothing with an 8 mm full width at half maximum Gaussian kernel. A general linear model with two conditions (decision phase and result phase) was estimated. No subject had to be excluded. The contrast images calculated for individual subjects were entered into a second level or random effects analysis [26]. We first calculated $t$-tests for the two conditions (decision/result) for all participants as one group and in a second step analysed each group (patients/controls) separately in the same way. In a third step, we calculated two-sample $t$-tests between the two groups for both conditions. We then implemented a $2 \times 2$ repeated measures ANOVA using a flexible factorial design with the factors 'group' (patients/controls) and 'condition' (decision/result) to calculate main effects of group and condition as well as the interaction effect of group and condition. The resultant statistical parametric maps were thresholded using an FWE-corrected $p$-value threshold less than 0.001, reporting clusters greater than 20 voxels ($k = 20$) only. Anatomical structures of cluster maxima were labelled in Talairach space using the Talairach Client (http://www.talairach.org/client.html).

Based on the previous research, we *a priori* selected the following regions of interest (ROIs) for further analyses: superior, middle and inferior frontal gyri, medial and lateral orbital gyri, posterior orbital gyrus, straight gyrus, anterior cingulate gyrus, amygdala, thalamus and caudate nucleus. We used adult brain maximum probability maps (© Copyright Imperial College of Science, Technology and Medicine 2007. All rights reserved) to obtain the ROIs [27].

In a subsequent step, we computed the % of BOLD signal change extracted from beta images from significant voxels from the second-level analysis within an 8 mm sphere surrounding the activation peak and calculated correlations with all other variables collected during the study.

### 2.5.2. Neuropsychological and questionnaire data

The statistical analysis was performed with SPSS, v. 20 and involved two-tailed, non-parametric testing (Wilcoxon test, $\chi^2$-test), as well as Spearman's rho correlations. We corrected for multiple testing using the Bonferroni–Holm procedure and chose a significance level of less than 0.05.

# 3. Results

## 3.1. Behavioural data, neuropsychological variables and questionnaires

We included 15 FLE patients and 15 healthy controls. Four patients (26.67%) were seizure free, while the remaining eleven (73.33%) had a median monthly seizure frequency of 4 during a three-month period preceding the experiment. MRI was normal in nine patients (60%) and showed frontal lobe lesions—cavernoma (one patient), focal cortical dysplasia (one patient), diffuse axonal trauma (one patient), and post-haematoma scars (two patients)—in the remaining five patients (33.3%). The remaining patient had undergone skull (but not brain) surgery in childhood for craniostenosis and the current structural MRI showed no obvious pathological changes. Three patients were on monotherapy and

**Table 2.** Summary of data collected on both participant groups of our study.

| variables—mean ± s.d. | patients (n = 15) | controls (n = 15) | z-score | p-value |
|---|---|---|---|---|
| age (years) | 36.0 ± 8.10 | 34.07 ± 6.05 | −0.769 | 0.442 |
| sex | 8 females | 8 females | n.a. | n.a. |
| **game behaviour (no. of cooperative choices)** | 70.20 ± 37.24 | 32.67 ± 17.56 | −3.298 | p = 0.001 |
| **game behaviour—game 1** | 20.73 ± 7.35 | 11.60 ± 5.42 | z = −2.501[a] | p = 0.012[a] |
| **game behaviour—game 4** | 15.73 ± 10.69 | 5.13 ± 5.49 | z = −2.759[a] | p = 0.006[a] |
| trustworthiness (sum of all evaluations on a Likert scale 1 to 7) | 34.15 ± 7.73 | 31.20 ± 5.78 | −1.131 | 0.258 |
| trust in doctors and health system | 21.33 ± 5.16 | 24.73 ± 6.09 | −1.713 | 0.087 |
| **theory of mind (REMT − no. of correct answers)** | 21.53 ± 4.84 | 27.00 ± 3.85 | −3.298 | 0.002 |
| **memory for faces (Wechsler subset, no. of correct answers)** | 71.50 ± 8.56 | 79.57 ± 6.82 | −2.454 | 0.012 |
| working memory for numbers (Wechsler subset, no. of correct answers) | 16.43 ± 3.5 | 18.21 ± 3.87 | −1.248 | 0.258 |
| **trail making A (s)** | 33.36 ± 12.40 | 22.00 ± 6.26 | −2.920 | 0.003 |
| **trail making B (s)** | 110.64 ± 70.15 | 58.29 ± 23.75 | −2.963 | 0.002 |

[a]The first value in this cell relates to the comparison between game 1 and 4 in the patient group, the second value in the same cell relates to same comparison in the control group; variables in bold differ significantly between groups; Morisky adherence scale: high values implying low therapy adherence; BMQ, Beliefs about medicines questionnaire: higher values represent higher belief in necessity, concerns and harms, or medication overuse/over-prescription by physicians; SATMED-Q: higher scores represent higher treatment satisfaction; n.a., not applicable.

12 patients were under polytherapy (mean number ± s.d. of anti-epileptic drugs: 2.33 ± 0.98). Mean ± s.d. subjective (Morisky adherence scale) and objective (pill counts) measures of treatment adherence during a six-month period were 1.73 ± 0.88 (range: 0–8, lower number indicates higher adherence) and 7.00 ± 12.45 (number of prescribed pills not taken), indicating moderate level of adherence. Patients also showed high belief of treatment necessity (BMQ mean ± s.d.—necessity 21.13 ± 6.30, range: 5–25), average level of concerns and negative views regarding therapy (BMQ: concerns and harms 24.87 ± 9.36, range: 9–45; overuse 7.47 ± 3.27, range: 4–20) and an average treatment satisfaction (SATMED-Q mean ± s.d.: 65.36 ± 7.88, range: 17–85). Questionnaires capturing beliefs about medicines ($p = 0.256$), treatment satisfaction ($p = 0.776$), or adherence ($p = 0.056$) did not correlate significantly with therapy adherence measured through pill counts, although the Morisky adherence scale was close to being significant ($p = 0.056$).

For all demographic, neuropsychological and neuroeconomic data collected in this study on both patient and control groups, please refer to table 2. In summary, there was no significant difference between the patient and control groups regarding age, gender, handedness, education, professional status, trust in doctors as captured by the respective questionnaire [19], trustworthiness of game opponents, or working memory for numbers. By contrast, patients demonstrated statistically significant worse performance than controls in mentalizing measured through the Reading eyes in the mind test, memory for faces, psychomotor speed during Trail making test A, and mental flexibility performance during Trail making test B.

Furthermore, there was also a significant difference in the total number of cooperative choices during the PDG, indicating higher cooperation in patients than in controls. The numbers of cooperative choices between games 1 and 4 differed significantly in both groups, indicating that cooperation decreased significantly between the first and the last game of the experiment in both groups (figure 2).

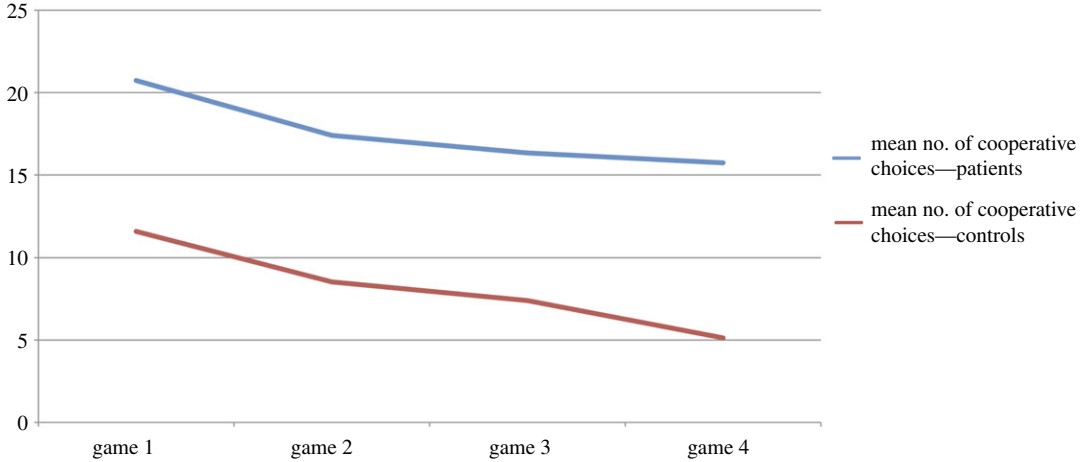

**Figure 2.** Mean numbers of cooperative choices of the patient and control groups in the four prisoners' dilemma games.

Mentalizing correlated strongly with memory for faces ($r = 0.715$; $p = 0.000$) and with trust in doctors and the health system [19] ($r = 0.437$; $p = 0.026$), showing that participants with high mentalizing abilities also better memorized faces and had higher trust in healthcare. Taken together these data supported a positive correlation between several pro-social cognition and behaviour variables.

There was a significant positive correlation between cooperative choices in the PDG and missed medication intakes with high values representing low therapy adherence ($r = 0.686$; $p = 0.010$), indicating higher cooperative behaviour in patients with low therapy adherence.

## 3.2. Imaging results (see tables 3, 4 and 5)

### 3.2.1. Voxel-based whole brain analysis

*Brain activation during decision-making.* During the decision-making whether to cooperate or defect when playing the PDG, combined event-related fMRI analysis of data from all subjects (one sample $t$-test) showed significantly activated clusters in right inferior parietal lobule, left precuneus, left lingual gyrus, left insula, the middle and inferior frontal gyri bilaterally, left superior frontal gyrus, as well as the right anterior lobe of the cerebellum (figure 3). In controls, significant activation was observed in the right fusiform gyrus, right superior frontal gyrus, left insula and left superior temporal gyrus, while in patients, significant activation was observed in the left superior frontal gyrus, left precuneus, and left insula.

*Brain activation during result phase.* During the perception of the results of the game, combined analysis of data from all subjects (one sample $t$-test) showed activation in the inferior parietal lobules bilaterally and the right middle temporal gyrus—these results are visualized in figure 3 as well. In controls, significant activation was observed in the left middle frontal gyrus, inferior parietal lobules bilaterally, right inferior temporal gyrus, and left cingulate gyrus. In patients, significant activation was observed in both inferior parietal lobules, left superior temporal gyrus, right middle frontal gyrus, right thalamus, left insula and right precuneus.

*Individual group comparisons for both conditions.* Two-sample $t$-tests between the patient and control groups did not yield any significant results for both conditions.

*Integrated comparison across groups and conditions.* In the full factorial analysis, the main group effect ($F$-test) showed significant clusters in the right and left medial prefrontal cortex—Brodmann area 10 (figure 4). In the main effect of condition (decision-making versus result phase of the game), the following clusters were significant: right anterior lobe of the cerebellum, left precentral gyrus, right inferior frontal gyrus, right insula, left middle temporal gyrus and left superior frontal gyrus. There was no significant interaction between group and condition.

### 3.2.2. Region of interest analysis

ROI analysis ($F$-test) showed differences in activation between the patient and control groups in the right and left superior frontal gyrus, while the $t$-test analyses showed no significant differences. The

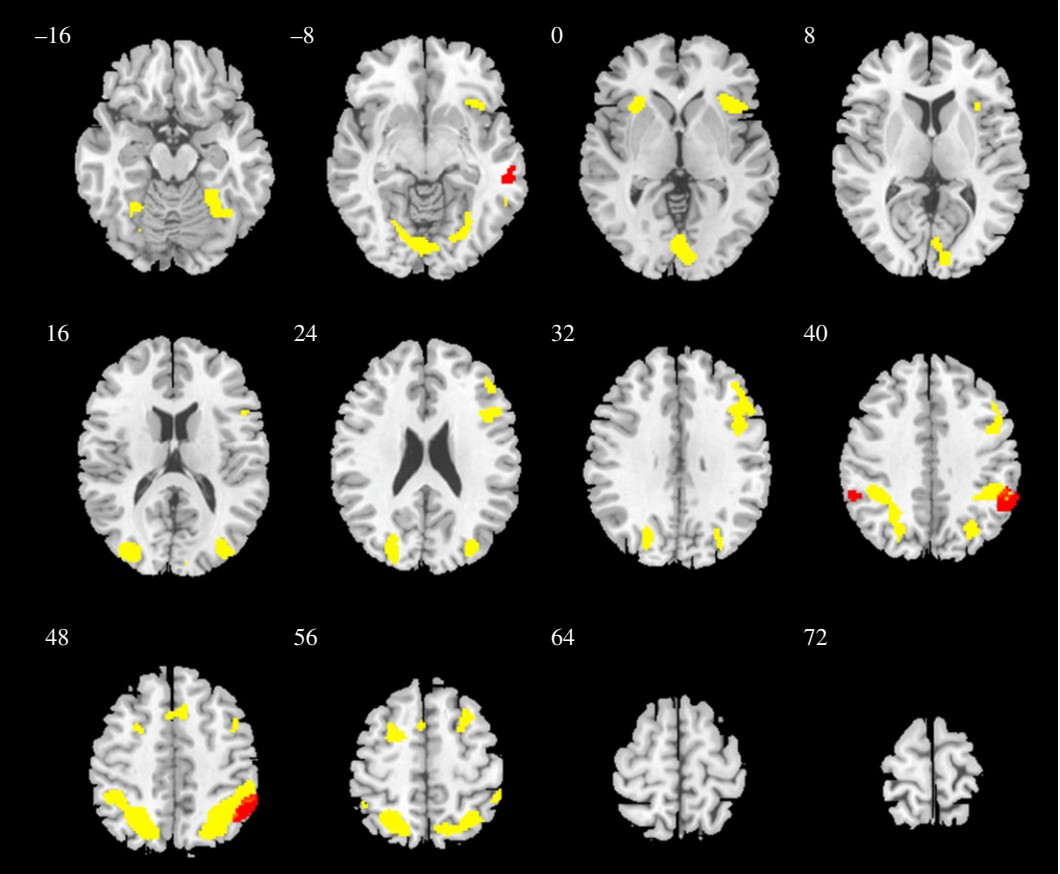

**Figure 3.** Visualization of brain activation of all participants (according to one sample $t$-test results reported in table 3) during decision-making (yellow) and result perception (red)—thresholded at $p < 0.001$, FWE-corrected, $k = 20$.

mean % of BOLD signal changes in these ROIs in the result condition showed a significant negative correlation with initial trust ($r = -0.448$; $p = 0.017$), indicating that participants with high initial trust showed lower signal change in these brain areas during result perception. The mean % of BOLD signal changes in the choice condition also correlated significantly with differences between the initial and final trust ($r = -0.618$; $p = 0.000$; first ROI and $r = -0.545$; $p = 0.003$ for the second ROI), meaning that participants with a high decrease of trust from the beginning to the end of the games showed lower signal changes while making their choices in the game.

Another significant correlation was detected between therapy adherence and the % BOLD signal change in the choice condition ($r = 0.565$; $p = 0.044$ for the first ROI) showing that participants with low therapy adherence had higher signal changes in this brain area.

## 4. Discussion

In this study, we first confirmed that mentalizing is impaired in patients with FLE, as previously established [28]. These abnormalities correlated with an altered memory for faces, a previously unreported finding in FLE, which may shed light on the mechanisms underlying dysfunction in social behaviour. Interestingly, there is evidence of frontal lobe contributions to memory for faces [29]. Furthermore, mentalizing abilities correlated with the results of the trust in healthcare questionnaire implying an overall good correlation of pro-social cognition variables in our study.

One of the goals of this study was to find out whether deficits in social cognition in FLE patients could lead to differences in social behaviour that can be operationalized. In fact, we showed that FLE patients behave differently from healthy controls in the PDG, but in a counterintuitive way, since they cooperated more than controls. While this finding was unexpected, we observed a frequent pattern of evolving cooperation during the game in both the patients' and controls' groups, with higher rates of

**Table 3.** Summary of analysis of all participants. FWE, $p = 0.001$.

| hemisphere | lobe | cluster covering | Brodmann area | p-value | K size | T score | x | y | z |
|---|---|---|---|---|---|---|---|---|---|
| *decision-making* | | | | | | | | | |
| right | parietal | inferior parietal lobule | BA 40 | 0.000 | 1720 | 13.10 | 48 | −38 | 46 |
| left | parietal | precuneus | BA 7 | 0.000 | 1748 | 12.95 | −20 | −66 | 52 |
| left | occipital | lingual gyrus | BA 18 | 0.000 | 817 | 10.63 | −4 | −82 | −6 |
| left | frontal | middle frontal gyrus | BA 6 | 0.000 | 194 | 10.58 | −28 | 8 | 52 |
| left | insula | insula | BA 13 | 0.000 | 130 | 10.55 | −30 | 22 | 0 |
| right | frontal | middle frontal gyrus | BA 9 | 0.000 | 733 | 10.43 | 46 | 34 | 30 |
| right | cerebellum anterior lobe | culmen | — | 0.000 | 368 | 10.32 | 26 | −48 | −14 |
| right | frontal | inferior frontal gyrus | BA 47 | 0.000 | 300 | 10.27 | 42 | 20 | −2 |
| right | frontal | inferior frontal gyrus | BA 6 | 0.000 | 105 | 9.90 | 28 | 16 | 56 |
| left | frontal | superior frontal gyrus | BA 6 | 0.000 | 166 | 9.62 | −6 | 12 | 52 |
| right | temporal | middle temporal gyrus | BA 37 | 0.000 | 40 | 9.36 | 56 | −52 | −12 |
| *results* | | | | | | | | | |
| right | parietal | inferior parietal lobule | BA 40 | 0.000 | 343 | 10.57 | 52 | −50 | 46 |
| right | temporal | middle temporal gyrus | BA 21 | 0.000 | 74 | 8.72 | 58 | −34 | −8 |
| left | parietal | inferior parietal lobule | BA 40 | 0.000 | 36 | 8.14 | −58 | −42 | 40 |

**Table 4.** Summary of per group analyses. FWE, $p = 0.001$.

| hemisphere | lobe | cluster covering | Brodmann area | p-value | K size | T score | x | y | z |
|---|---|---|---|---|---|---|---|---|---|
| PATIENTS | | | | | | | | | |
| *decision-making* | | | | | | | | | |
| right | occipital | lingual gyrus | BA 18 | 0.000 | 11 406 | 10.64 | 24 | −74 | −10 |
| left | frontal | superior frontal gyrus | BA 6 | 0.000 | 8491 | 10.28 | −16 | 0 | 64 |
| left | subcortical insula | claustrum insula | — | 0.006 | 456 | 7.63 | −28 | 22 | −2 |
| *results* | | | | | | | | | |
| left | parietal | inferior parietal lobule | BA 40 | 0.000 | 4211 | 12.48 | −60 | −42 | 38 |
| right | parietal | inferior parietal lobule | BA 40 | 0.000 | 5094 | 11.53 | 58 | −44 | 44 |
| right | frontal | middle frontal gyrus | BA 46 | 0.000 | 2994 | 10.12 | 42 | 48 | 14 |
| left | cerebellum | anterior lobe | — | 0.003 | 439 | 6.91 | −2 | −54 | −4 |
| right | parietal | precuneus | BA 7 | 0.016 | 304 | 6.35 | 4 | −78 | 46 |
| CONTROLS | | | | | | | | | |
| *decision-making* | | | | | | | | | |
| right | temporal occipital | fusiform gyrus | BA 37 | 0.000 | 17309 | 13.94 | 54 | −54 | −16 |
| right | frontal | superior frontal gyrus | BA 9 | 0.000 | 9140 | 11.15 | 42 | 36 | 28 |
| left | insula | insula | BA 13 | 0.000 | 756 | 8.1 | −34 | 18 | 8 |
| left | temporal | superior temporal gyrus | BA 22 | 0.020 | 238 | 7.69 | −50 | −48 | 14 |
| *results* | | | | | | | | | |
| left | frontal | middle frontal gyrus | BA 10 | 0.000 | 67860 | 9.12 | −34 | 48 | −2 |
| left | parietal | inferior parietal lobule | BA 40 | 0.000 | 3234 | 7.88 | −46 | −46 | 56 |
| right | parietal | inferior parietal lobule | BA 40 | 0.000 | 1086 | 7.11 | 52 | −50 | 46 |
| right | temporal | inferior temporal gyrus | BA 20 | 0.000 | 596 | 7.08 | 58 | −40 | 16 |
| left | frontal | cingulate gyrus | BA 24 | 0.046 | 219 | 5.41 | −2 | −14 | 38 |

**Table 5.** Analysis of group differences and ROI.

| hemisphere | lobe | cluster covering | Brodmann area | p-value | K size | F score | x | y | z |
|---|---|---|---|---|---|---|---|---|---|
| *main effect of group* | | | | | | | | | |
| left | frontal | medial frontal gyrus | BA 10 | 0.034 | 214 | 22.35 | 8 | 62 | 14 |
| *main effect of condition* | | | | | | | | | |
| right | cerebellum | anterior lobe | — | 0.000 | 11343 | 108.49 | 26 | −50 | −12 |
| left | frontal | precentral gyrus | BA 9 | 0.000 | 1439 | 81.83 | −36 | 4 | 34 |
| right | frontal | inferior frontal gyrus | BA 9 | 0.000 | 1770 | 64.81 | 42 | 10 | 26 |
| right | insula | insula | BA 13 | 0.002 | 414 | 56.86 | 32 | 22 | −2 |
| left | temporal | middle temporal gyrus | BA 21 | 0.000 | 6458 | 52.55 | −56 | −22 | −14 |
| right | insula | insula | BA 13 | 0.000 | 4254 | 50.94 | 42 | −12 | 6 |
| left | frontal | cingulate gyrus | BA 24 | 0.000 | 1827 | 40.86 | −4 | −14 | 40 |
| *ROI-main effect of group* | | | | | | | | | |
| right | frontal | superior frontal gyrus | BA 10 | 0.007 | 219 | 31.57 | 14 | 60 | 14 |
| left | frontal | middle frontal gyrus | BA 10 | 0.046 | 124 | 19.57 | −32 | 54 | 8 |

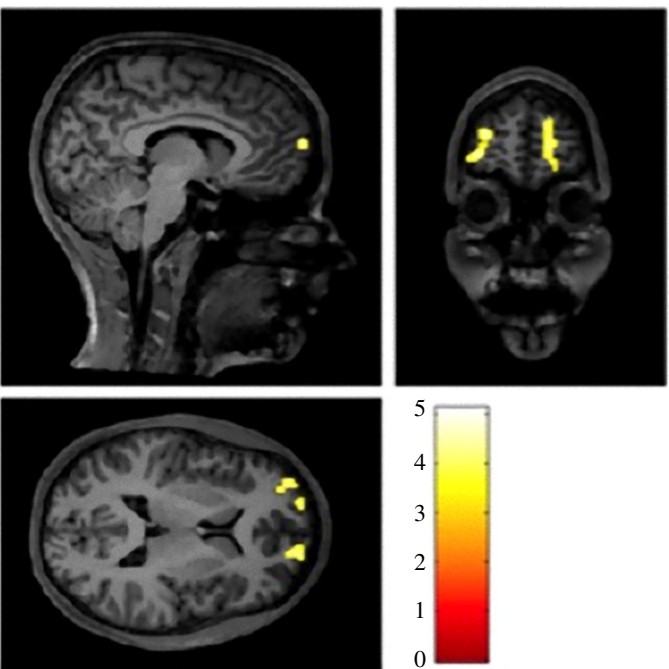

**Figure 4.** Difference in activation between the patient and control groups (main effect of groups, *F*-test) in the whole-brain analysis—thresholded at $p < 0.001$, FWE-corrected, $k = 20$.

cooperation in early game phases and less cooperative behaviour in the later ones [30]. This strategy is consistent with the objective of maximizing profit. Indeed, future possible interactions encourage people to cooperate in the early phases of the game, with the hope to initiate a mutually cooperative relationship. If not reciprocated, strategy will later shift to defection in players who want to maximize their own profit [31].

In this context, higher cooperation in patients might result from impaired negative feedback when cooperative behaviour is not reciprocated, especially because the ability to shift behaviour from cooperation to defection in such a case is dependent on frontal lobe functions [32]. Alternatively, patients' behaviour could be interpreted as a preference for higher delayed rewards since higher cooperation leads to higher future profits [33]. Although evidence on time preference for rewards (also referred to as 'delay discounting') in neurological patients is scarce, altered delay discounting as another behavioural symptom of neurological disorders was reported before [34].

Prior functional brain imaging studies investigating cooperative behaviour through the PDG found, in essence, three brain networks to be active during the game: several frontal brain areas, especially the medial prefrontal cortex, as well as reward and limbic brain regions [5] reflecting the cognitive functions necessary to successfully play the game, such as decision-making and reward-based learning. Our analyses of brain activation of all participants showed a similar pattern of brain areas to be active during the game. However, when looking at the patient and control group separately, controls, but not patients showed significant activation in several temporal regions during decision-making whether to cooperate or not. Interestingly, those regions have been implicated in both Theory-of-Mind and facial processing tasks (e.g. [35]) and thus further substantiate our behavioural results of higher mentalizing and face memory abilities in the control group. During the perception of the results during the game, the cingulate gyrus, a region thought to be active in situations of cognitive conflict [36], was significantly activated in controls, but not in patients. When opponents in an economic game do not reciprocate benevolent behaviour, this is perceived as a conflict between economic self-interest (i.e. rationality) and fairness considerations [37]. Hypothetically, patients might have not perceived cognitive conflicts to the same extent as controls in such game results, which might also have led to higher cooperative behaviour.

When comparing our functional imaging results between the patient and control groups on both whole-brain and ROI level, it becomes apparent that the activation of the medial prefrontal cortex (MPFC) differed significantly between the two groups during the game. The MPFC plays a pivotal role in social behaviour and decision-making [32,35], which is reflected in our results of correlating

MPFC activation changes and face trustworthiness evaluations, although those have been based upon a low number of data points only. Therefore, a difference in MPFC activation between two groups showing such divergent cooperative behaviour seems plausible. Our fMRI findings do not allow us to infer the directionality of the observed difference in activation, potentially because of either the difference being based on both conditions rather than differences in activation during the individual conditions, the *t*-test comparisons between groups being underpowered, and/or the difference in activation between the groups not being sufficiently strong to pass the set threshold in the *t*-test analyses. Although a pathology-driven lower activation might seem more plausible, a compensatory higher activation is also possible [38]. Further imaging studies are needed to address this question.

Another goal of our study was to investigate possible links between cooperative behaviour and therapy adherence in FLE. Our results show that cooperative behaviour in the PDG correlated negatively with therapy adherence in patients (high cooperative behaviour correlating with low therapy adherence). Even though we could not find any previous literature on the relationship between cooperative behaviour and therapy adherence, there is some evidence that decision-making, in general, has effects on therapy adherence with lower decision-making skills leading to lower adherence [39]. As we have outlined before, higher cooperation can be interpreted as lower social decision-making abilities in the context of our experiment. Thus, the inverse relation between therapy adherence and cooperative behaviour actually fits into the current scientific framework on the interaction of behaviour, decision-making and therapy adherence. Importantly, there was no correlation between questionnaire data about therapy adherence and pill counts in our study, reflecting the known difficulties in measuring adherence in epilepsy (e.g. [7]). Moreover, the MPFC activation during the game showed a correlation with therapy adherence reflecting the close link between social cognition and therapy adherence.

We acknowledge that our study design of comparing people with FLE to healthy controls does not allow distinguishing whether the study findings apply solely to FLE or to epilepsy in general. Looking at the published literature on other epilepsy syndromes with frontal lobe dysfunction, such as juvenile myoclonic epilepsy or genetic generalized epilepsies, it becomes apparent that patients with these types of epilepsies also show abnormalities in social functions such as Theory-of-Mind [40,41]. In our view, neuroeconomic methodology could help us to determine the behavioural consequences of these social cognitive impairments. Further studies comparing cooperative behaviour and therapy adherence between FLE and other epileptic disorders are therefore warranted.

# 5. Conclusion

To conclude, our results implicate that (i) social behaviour is affected by FLE and (ii) can be measured using neuroeconomic methods. Impaired social behaviour in FLE might (iii) be a consequence of differing MPFC activation and (iv) might play a role in low therapy adherence. This is important as therapy adherence is difficult to measure, especially in patients with epilepsy where classical tools of measurement (e.g. questionnaires or electronic devices) have been shown to be imprecise [7]. Integrating neuroeconomic testing of social behaviour into the neuropsychological testing routine could help to better understand therapy adherence of patients with epilepsies and consequently help to improve patient care through an identification of patients at risk.

Ethics. The study has been approved by the local ethics committee (CPP Sud Est II no. 2014-A01689-38) and competent authority (ANSM no. 141399B-31). All patients gave written informed consent. The study was registered under clinicaltrials.gov (NCT02441478).

Data accessibility. The datasets supporting this article have been uploaded as part of the electronic supplementary material.

Authors' contributions. A.J. has contributed to the conception and design of the study, the acquisition and analysis of data and writing the first draft of the manuscript and figures as well as coordinating the development of the final draft. C.C., P.R. and S.R. have contributed to the conception and design of the study, the acquisition and analysis of data and drafting a significant portion of the manuscript and figures. D.I. has contributed to the acquisition and analysis of data and drafting a significant portion of the manuscript and figures.

Competing interests. A.J. has moved to Biogen International GmbH, Zug, Switzerland since completion of this work. The other authors declare no competing interests.

Funding. This research was partly supported by a grant of the Austrian Societies for Neurology and Epileptology.

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
