## [Reviewer comments · Royal Society Open Science]

Review History

RSOS-180850.R0 (Original submission)

Review form: Reviewer 1

Is the manuscript scientifically sound in its present form?

Yes

Are the interpretations and conclusions justified by the results?

Yes

Is the language acceptable?

Yes

Is it clear how to access all supporting data?

Yes

Do you have any ethical concerns with this paper?

No

Have you any concerns about statistical analyses in this paper?

I do not feel qualified to assess the statistics

Recommendation?

Accept with minor revision (please list in comments)

Comments to the Author(s)

This is an interesting paper. Aims are original and clear, so are methods and results and discussion is thorough.

I have some comments and suggestions though.

1. To improve readability I suggest to shorten the methods. For example, sentences like “we subsequently checked whether subjects understood the information...” or “(executive/non executive/no profession....)” can be deleted without subtracting important information and making it easier to follow the paper.

An analogous consideration can be done for the long premise in the Result section in the Imaging part (“according to the analysis outlined...”)

2. In the results some data are replicated. I suggest that data referring only to the patients are reported only in the text and data comparing patients and controls are reported only in the table. Whatever the choice, data have to be reported once as duplicates worsen readability.

3. In the discussion section I don’t think that the sentence, “ In fact , there is evidence that neurological disorders like Parkinson’s disease can shift the time preference for rewards”, formulated as such, is pertinent.

4. The abstract does not mention the imaging part in the design of the study and results. I suppose this is an involuntary omission.

Finally I did not understand whether the computer-generated strategy of “player 2” is designed to defect rather than cooperate, thus pushing the subject to change strategy shifting towards defecting over time.

Review form: Reviewer 2**Is the manuscript scientifically sound in its present form?**

Yes

Are the interpretations and conclusions justified by the results?

No

Is the language acceptable?

Yes

Is it clear how to access all supporting data?

Yes

Do you have any ethical concerns with this paper?

No

Have you any concerns about statistical analyses in this paper?

No

Recommendation?

Reject

Comments to the Author(s)

This is a well written, clearly presented study looking at the differences between people with frontal lobe epilepsy and healthy controls on a prisoner's dilemma task, and the ways in which their responses have an impact on medication compliance. This is an important area of study and the study presents an innovation way of looking at this, with the added potential of clinically relevant findings.

It has a number of limitations, common to all studies of clinical populations, not least the difficulties in accurately measuring compliance with treatment. These are acknowledged by the authors. However there are two larger methodological shortcomings that significantly limit the conclusions that can be drawn from the data.

Firstly, no epilepsy control group was used. It is therefore unclear whether the study findings represent FLE or the impact of epilepsy in general or indeed antiepileptic medications. This is particularly important given the counterintuitive results which indicate that people with FLE cooperate more than controls - could this be due to the social impact of living with a challenging, stigmatising condition? Is this behaviour more common in all people with epilepsy?

Secondly post operative patients appear to have been included in the FLE group, These patients will obviously have very different fMRI activations than non-surgical samples and there is increasing evidence in the fMRI literature that surgery changes patterns of activation. This adds yet another important confounding variable in this already small sample.

Decision letter (RSOS-180850.R0)

12-Oct-2018

Dear Dr Javor,

The editors assigned to your paper ("Social cognition, behaviour, and therapy adherence in frontal lobe epilepsy: A study combining neuroeconomic and neuropsychological methods") have now received comments from reviewers. We would like you to revise your paper in accordance with the referee and Associate Editor suggestions which can be found below (not including confidential reports to the Editor). Please note this decision does not guarantee eventual acceptance.

In particular, please add a section to the discussion addressing the limitations raised by reviewer 2, and edit the manuscript as suggested by reviewer 1.

Please submit a copy of your revised paper before 04-Nov-2018. Please note that the revision deadline will expire at 00.00am on this date. If we do not hear from you within this time then it will be assumed that the paper has been withdrawn. In exceptional circumstances, extensions may be possible if agreed with the Editorial Office in advance. We do not allow multiple rounds of revision so we urge you to make every effort to fully address all of the comments at this stage. If deemed necessary by the Editors, your manuscript will be sent back to one or more of the original reviewers for assessment. If the original reviewers are not available, we may invite new reviewers.

To revise your manuscript, log into <http://mc.manuscriptcentral.com/rsos> and enter your

Author Centre, where you will find your manuscript title listed under "Manuscripts with Decisions." Under "Actions," click on "Create a Revision." Your manuscript number has been appended to denote a revision. Revise your manuscript and upload a new version through your Author Centre.

- Data accessibility

If you wish to submit your supporting data or code to Dryad (<http://datadryad.org/>), or modify your current submission to dryad, please use the following link:
<http://datadryad.org/submit?journalID=RSOS&manu=RSOS-180850>

- Competing interests

- Authors' contributions

AB carried out the molecular lab work, participated in data analysis, carried out sequence alignments, participated in the design of the study and drafted the manuscript; CD carried out the statistical analyses; EF collected field data; GH conceived of the study, designed the study,

coordinated the study and helped draft the manuscript. All authors gave final approval for publication.

- Acknowledgements

- Funding statement

Please note that Royal Society Open Science charge article processing charges for all new submissions that are accepted for publication. Charges will also apply to papers transferred to Royal Society Open Science from other Royal Society Publishing journals, as well as papers submitted as part of our collaboration with the Royal Society of Chemistry (<http://rsos.royalsocietypublishing.org/chemistry>). If your manuscript is newly submitted and subsequently accepted for publication, you will be asked to pay the article processing charge, unless you request a waiver and this is approved by Royal Society Publishing. You can find out more about the charges at <http://rsos.royalsocietypublishing.org/page/charges>. Should you have any queries, please contact openscience@royalsociety.org.

on behalf of Dr Anastasia Christakou (Associate Editor) and Prof. Antonia Hamilton (Subject Editor)
openscience@royalsociety.org

Comments to Author:

Reviewers' Comments to Author:

Reviewer: 1

Comments to the Author(s)

This is an interesting paper. Aims are original and clear, so are methods and results and discussion is thorough.

I have some comments and suggestions though.

1. To improve readability I suggest to shorten the methods. For example, sentences like “we subsequently checked whether subjects understood the information...” or “(executive/non executive/no profession...)” can be deleted without subtracting important information and making it easier to follow the paper.

An analogous consideration can be done for the long premise in the Result section in the Imaging part (“according to the analysis outlined...”)

2. In the results some data are replicated. I suggest that data referring only to the patients are reported only in the text and data comparing patients and controls are reported only in the table. Whatever the choice, data have to be reported once as duplicates worsen readability.

3. In the discussion section I don't think that the sentence, " In fact , there is evidence that neurological disorders like Parkinson's disease can shift the time preference for rewards", formulated as such, is pertinent.

4. The abstract does not mention the imaging part in the design of the study and results. I suppose this is an involuntary omission.

Finally I did not understand whether the computer-generated strategy of "player 2" is designed to defect rather than cooperate, thus pushing the subject to change strategy shifting towards defecting over time.

Reviewer: 2

Comments to the Author(s)

This is a well written, clearly presented study looking at the differences between people with frontal lobe epilepsy and healthy controls on a prisoner's dilemma task, and the ways in which their responses have an impact on medication compliance. This is an important area of study and the study presents an innovation way of looking at this, with the added potential of clinically relevant findings.

It has a number of limitations, common to all studies of clinical populations, not least the difficulties in accurately measuring compliance with treatment. These are acknowledged by the authors However there are two larger methodological shortcomings that significantly limit the conclusions that can be drawn from the data.

Firstly, no epilepsy control group was used. It is therefore unclear whether the study findings represent FLE or the impact of epilepsy in general or indeed antiepileptic medications. This is particularly important given the counterintuitive results which indicate that people with FLE cooperate more than controls - could this be due to the social impact of living with a challenging, stigmatising condition? Is this behaviour more common in all people with epilepsy?

Secondly post operative patients appear to have been included in the FLE group, These patients will obviously have very different fMRI activations than non-surgical samples and there is increasing evidence in the fMRI literature that surgery changes patterns of activation. This adds yet another important confounding variable in this already small sample.

Author's Response to Decision Letter for (RSOS-180850.R0)

See Appendix A.

RSOS-180850.R1 (Revision)

Review form: Reviewer 1

Is the manuscript scientifically sound in its present form?

Yes

Are the interpretations and conclusions justified by the results?

Yes

Is the language acceptable?

Yes

Is it clear how to access all supporting data?

Yes

Do you have any ethical concerns with this paper?

No

Have you any concerns about statistical analyses in this paper?

I do not feel qualified to assess the statistics

Recommendation?

Accept as is

Comments to the Author(s)

My remarks were adequately addressed.

Review form: Reviewer 3

Is the manuscript scientifically sound in its present form?

Yes

Are the interpretations and conclusions justified by the results?

No

Is the language acceptable?

Yes

Is it clear how to access all supporting data?

Not Applicable

Do you have any ethical concerns with this paper?

No

Have you any concerns about statistical analyses in this paper?

Yes

Recommendation?

Major revision is needed (please make suggestions in comments)

Comments to the Author(s)

Many thanks for the opportunity to review this paper. The authors analysed a sample of 15 patients with frontal lobe epilepsy and 15 healthy controls using neuropsychometry, social cognition testing and an fMRI dilemma game task. Patients exhibited altered activation of the prefrontal cortex, which related to performance in social cognition tests as well as compliance with anti-epileptic drug therapy.

The paper is novel, very interesting, and significantly adds to the literature of social cognition in epilepsy. It will definitely be of interest to the readership of Royal Society Open Science.

I have a number of observations as follows – though most are minor, please note that there are some major concerns regarding the statistical analysis of imaging data, which should be addressed thoroughly.

Abstract

Could the authors specify the number of subjects analysed in the abstract already?

Methods

1. "Subjects" section – could the authors be more specific as to what was the cut-off for 'frontal lobe lesions affecting large cortical areas' to grant exclusion of a participant?
2. "Paradigm" section, second line": please change "insure" with "ensure".
3. Regarding the cognitive abilities tested by the Trail Making Test A and B-A, the authors should be more specific, as the latter does not merely assess attention. TMT B-A is a measure of 'mental flexibility', which is an executive function, whereas TMT-A more appropriately relates to psychomotor speed. This denominations, consequently, should be adopted in the subsequent manuscript sections too.

Results

1. In line with suggestions of previous reviewer, I would recommend further streamlining of the main text. For instance, in the "Behavioural data, neuropsychological variables and questionnaires" section of the manuscript, all z scores and p values for group comparisons could be moved to Table 2. This would also enhance the interpretability of the results reported in the table.
2. When describing first-level imaging data analysis, the authors should specify the type of analysis done (supposedly event-related fMRI) for the convenience of the reader.
3. It would be helpful to provide a figure from a one-sample showing the main effects of task across study groups, thus highlighting the task-associated whole-brain activation map (i.e., an image related to the findings detailed in Table 3).
4. I have concerns on how imaging results were reported with regards to group comparisons. In a 2x2 factorial SPM design, the 'Main effect' of group contrast is helpful to identify areas of group differences in either direction (FLE > CTR or FLE < CTR), is associated with an F statistic, but does not provide information regarding the directionality of effects. The latter is provided by t-contrasts, which are not reported in the manuscript (there are only F-scores in Table 5). How can the authors conclude that activation of the superior frontal gyrus is higher in patients (and not lower), when this was not formally tested? This is a relevant shortcoming and has to be addressed. Appropriate t-contrasts need be carried out both for whole-brain and ROI-based group comparisons.
5. Linked with the above comment – Figure 3 (group comparisons) should be re-done according to what group comparisons actually show.
6. Always with regards to group comparisons – can the authors confirm that, in the batch system, the following parameter were chosen for the factorial design: (a) Group - Independence: "Yes", but (b) for Condition (decision/result) → Independence should be set to "No" – as this is a case of repeated measures design.
7. I find the description of the correlation between imaging and behavioural measures a bit confusing. First, it is unclear to me what BOLD % signal change the authors are referring to – was it for the 'decision', for the 'result' condition, or for an average of both? Second, the whole phrasing of 'activation change' is misleading – either 'signal change' or 'activation/deactivation' should be used to refer to a change in % BOLD, activation change presumes that there's a dynamic process taking place during the task, which is not the case here.

Discussion

1. Depending on the actual directionality of the group contrast as discussed in the above comment, the whole section addressing imaging findings (page 13, second paragraph, continuing on the following page) may need to be reformulated
2. In the last paragraph, it would be helpful to put the findings in a broader context, e.g. by mentioning briefly the current state of the art re: social cognition impairment in other epilepsy syndromes with frontal lobe dysfunction, such as JME/IGE (see Giorgi et al., *Epilepsy Research* 2016; Stewart et al., *Seizure* 2018).

Decision letter (RSOS-180850.R1)

07-May-2019

Dear Dr Javor:

Manuscript ID RSOS-180850.R1 entitled "Social cognition, behaviour, and therapy adherence in frontal lobe epilepsy: A study combining neuroeconomic and neuropsychological methods" which you submitted to Royal Society Open Science, has been reviewed. The comments of the reviewer(s) are included at the bottom of this letter.

Please submit a copy of your revised paper before 30-May-2019. Please note that the revision deadline will expire at 00.00am on this date. If we do not hear from you within this time then it will be assumed that the paper has been withdrawn. In exceptional circumstances, extensions may be possible if agreed with the Editorial Office in advance. We do not allow multiple rounds of revision so we urge you to make every effort to fully address all of the comments at this stage. If deemed necessary by the Editors, your manuscript will be sent back to one or more of the original reviewers for assessment. If the original reviewers are not available we may invite new reviewers.

- Ethics statement

- Data accessibility

- Competing interests

- Authors' contributions

- Acknowledgements

- Funding statement

on behalf of Dr Anastasia Christakou (Associate Editor) and Antonia Hamilton (Subject Editor)
openscience@royalsociety.org

Associate Editor Comments to Author (Dr Anastasia Christakou):

Please accept our apologies for the unusual delay in completing the review of your manuscript: unfortunately, one of the original reviewers for your paper was not available, and we struggled to find an alternative referee to assess the revision. We have now received comments from an additional reviewer, who has a number of comments that we'd like you to address -- as a number of their concerns are major, we're offering a final opportunity of revision to resolve these problems (ordinarily, authors are entitled to one round of may revisions only). You should be aware that, if you're not able to persuade the reviewer that the manuscript is ready for acceptance, we will not be able to consider the paper further. We wish you success, and look forward to reading the revision in due course.

Associate Editor: 2

Comments to the Author:
(There are no comments.)

Reviewer comments to Author:

Reviewer: 1

Comments to the Author(s)

My remarks were adequately addressed.

Reviewer: 3

Comments to the Author(s)

Many thanks for the opportunity to review this paper. The authors analysed a sample of 15 patients with frontal lobe epilepsy and 15 healthy controls using neuropsychometry, social cognition testing and an fMRI dilemma game task. Patients exhibited altered activation of the prefrontal cortex, which related to performance in social cognition tests as well as compliance with anti-epileptic drug therapy.

The paper is novel, very interesting, and significantly adds to the literature of social cognition in epilepsy. It will definitely be of interest to the readership of Royal Society Open Science.

I have a number of observations as follows – though most are minor, please note that there are some major concerns regarding the statistical analysis of imaging data, which should be addressed thoroughly.

Abstract

Could the authors specify the number of subjects analysed in the abstract already?

Methods

1. "Subjects" section – could the authors be more specific as to what was the cut-off for 'frontal lobe lesions affecting large cortical areas' to grant exclusion of a participant?
2. "Paradigm" section, second line": please change "insure" with "ensure".
3. Regarding the cognitive abilities tested by the Trail Making Test A and B-A, the authors should be more specific, as the latter does not merely assess attention. TMT B-A is a measure of 'mental flexibility', which is an executive function, whereas TMT-A more appropriately relates to

psychomotor speed. This denominations, consequently, should be adopted in the subsequent manuscript sections too.

Results

1. In line with suggestions of previous reviewer, I would recommend further streamlining of the main text. For instance, in the "Behavioural data, neuropsychological variables and questionnaires" section of the manuscript, all z scores and p values for group comparisons could be moved to Table 2. This would also enhance the interpretability of the results reported in the table.
2. When describing first-level imaging data analysis, the authors should specify the type of analysis done (supposedly event-related fMRI) for the convenience of the reader.
3. It would be helpful to provide a figure from a one-sample showing the main effects of task across study groups, thus highlighting the task-associated whole-brain activation map (i.e., an image related to the findings detailed in Table 3).
4. I have concerns on how imaging results were reported with regards to group comparisons. In a 2x2 factorial SPM design, the 'Main effect' of group contrast is helpful to identify areas of group differences in either direction (FLE > CTR or FLE < CTR), is associated with an F statistic, but does not provide information regarding the directionality of effects. The latter is provided by t-contrasts, which are not reported in the manuscript (there are only F-scores in Table 5). How can the authors conclude that activation of the superior frontal gyrus is higher in patients (and not lower), when this was not formally tested? This is a relevant shortcoming and has to be addressed. Appropriate t-contrasts need be carried out both for whole-brain and ROI-based group comparisons.
5. Linked with the above comment – Figure 3 (group comparisons) should be re-done according to what group comparisons actually show.
6. Always with regards to group comparisons – can the authors confirm that, in the batch system, the following parameter were chosen for the factorial design: (a) Group - Independence: "Yes", but (b) for Condition (decision/result) → Independence should be set to "No" – as this is a case of repeated measures design.
7. I find the description of the correlation between imaging and behavioural measures a bit confusing. First, it is unclear to me what BOLD % signal change the authors are referring to – was it for the 'decision', for the 'result' condition, or for an average of both? Second, the whole phrasing of 'activation change' is misleading – either 'signal change' or 'activation/deactivation' should be used to refer to a change in % BOLD, activation change presumes that there's a dynamic process taking place during the task, which is not the case here.

Discussion

1. Depending on the actual directionality of the group contrast as discussed in the above comment, the whole section addressing imaging findings (page 13, second paragraph, continuing on the following page) may need to be reformulated
2. In the last paragraph, it would be helpful to put the findings in a broader context, e.g. by mentioning briefly the current state of the art re: social cognition impairment in other epilepsy syndromes with frontal lobe dysfunction, such as JME/IGE (see Giorgi et al., *Epilepsy Research* 2016; Stewart et al., *Seizure* 2018).

Author's Response to Decision Letter for (RSOS-180850.R1)

See Appendix B.

RSOS-180850.R2 (Revision)

Review form: Reviewer 3

Is the manuscript scientifically sound in its present form?

Yes

Are the interpretations and conclusions justified by the results?

Yes

Is the language acceptable?

Yes

Do you have any ethical concerns with this paper?

No

Have you any concerns about statistical analyses in this paper?

No

Recommendation?

Accept with minor revision (please list in comments)

Comments to the Author(s)

Many thanks for the opportunity to re-review this paper.

I am generally satisfied as to how the authors addressed my previous comments, and I appreciate the restructuring of the discussion.

A few minor points:

1. For the newly added Figure 3: it would be appreciated if, in the figure legend, the authors could confirm whether the shown activation maps were thresholded at $p < 0.001$, uncorrected (in line, I presume, with what specified in the "Imaging data" paragraph of the Data Analysis section). For consistency, this piece of information should also be added to the legend of Figure 4.

2. In the main text and figure legend, "one group t test" should be changed to "one sample t test".

3. When discussing the full factorial analysis (results section named: "Integrated comparison across groups and conditions", before region of interest analysis): the authors should specify, after "the main group effect", that this was an F-test, to enhance clarity. Such clarification can easily go in brackets.

4. In their response to my point 4 of the Results section, the authors state that additional t test analyses were conducted as requested, both for whole-brain and at a ROI level, and that the latter showed no significant group differences.

However, in the paragraph named "Region of interest analysis" of the Imaging results section (page 10), the first sentence states "ROI-analysis also showed areas of higher activation in patients than in controls in the right and left superior frontal gyrus". This appears in contrast with what the authors state in their response – could the authors clarify? It should be amended to reflect the fact that there were no significant group differences, I believe.

5. Discussion, 3 line from the beginning – guess the authors mean “shed light” instead of “shade light” ?

Decision letter (RSOS-180850.R2)

05-Jul-2019

Dear Dr Javor:

On behalf of the Editors, I am pleased to inform you that your Manuscript RSOS-180850.R2 entitled "Social cognition, behaviour, and therapy adherence in frontal lobe epilepsy: A study combining neuroeconomic and neuropsychological methods" has been accepted for publication in Royal Society Open Science subject to minor revision in accordance with the referee suggestions. Please find the referees' comments at the end of this email.

The reviewers and Subject Editor have recommended publication, but also suggest some minor revisions to your manuscript. Therefore, I invite you to respond to the comments and revise your manuscript.

- Ethics statement

- Data accessibility

If you wish to submit your supporting data or code to Dryad (<http://datadryad.org/>), or modify your current submission to dryad, please use the following link:
<http://datadryad.org/submit?journalID=RSOS&manu=RSOS-180850.R2>

- Competing interests

- Authors' contributions

- Acknowledgements

- Funding statement

Because the schedule for publication is very tight, it is a condition of publication that you submit the revised version of your manuscript before 14-Jul-2019. Please note that the revision deadline will expire at 00.00am on this date. If you do not think you will be able to meet this date please let me know immediately.

- 1) A text file of the manuscript (tex, txt, rtf, docx or doc), references, tables (including captions) and figure captions. Do not upload a PDF as your "Main Document".
- 2) A separate electronic file of each figure (EPS or print-quality PDF preferred (either format should be produced directly from original creation package), or original software format)
- 3) Included a 100 word media summary of your paper when requested at submission. Please ensure you have entered correct contact details (email, institution and telephone) in your user account
- 4) Included the raw data to support the claims made in your paper. You can either include your data as electronic supplementary material or upload to a repository and include the relevant doi within your manuscript

5) All supplementary materials accompanying an accepted article will be treated as in their final form. Note that the Royal Society will neither edit nor typeset supplementary material and it will be hosted as provided. Please ensure that the supplementary material includes the paper details where possible (authors, article title, journal name).

on behalf of Dr Anastasia Christakou (Associate Editor) and Antonia Hamilton (Subject Editor)
openscience@royalsociety.org

Reviewer comments to Author:

Reviewer: 3

Comments to the Author(s)

Many thanks for the opportunity to re-review this paper.

I am generally satisfied as to how the authors addressed my previous comments, and I appreciate the restructuring of the discussion.

A few minor points:

1. For the newly added Figure 3: it would be appreciated if, in the figure legend, the authors could confirm whether the shown activation maps were thresholded at $p < 0.001$, uncorrected (in line, I presume, with what specified in the "Imaging data" paragraph of the Data Analysis section). For consistency, this piece of information should also be added to the legend of Figure 4.
2. In the main text and figure legend, "one group t test" should be changed to "one sample t test".
3. When discussing the full factorial analysis (results section named: "Integrated comparison across groups and conditions", before region of interest analysis): the authors should specify, after "the main group effect", that this was an F-test, to enhance clarity. Such clarification can easily go in brackets.
4. In their response to my point 4 of the Results section, the authors state that additional t test analyses were conducted as requested, both for whole-brain and at a ROI level, and that the latter showed no significant group differences.

However, in the paragraph named “Region of interest analysis” of the Imaging results section (page 10), the first sentence states “ROI-analysis also showed areas of higher activation in patients than in controls in the right and left superior frontal gyrus”. This appears in contrast with what the authors state in their response – could the authors clarify? It should be amended to reflect the fact that there were no significant group differences, I believe.

5. Discussion, 3 line from the beginning – guess the authors mean “shed light” instead of “shade light” ?

Author's Response to Decision Letter for (RSOS-180850.R2)

See Appendix C.

Decision letter (RSOS-180850.R3)

23-Jul-2019

Dear Dr Javor,

I am pleased to inform you that your manuscript entitled "Social cognition, behaviour, and therapy adherence in frontal lobe epilepsy: A study combining neuroeconomic and neuropsychological methods" is now accepted for publication in Royal Society Open Science.

on behalf of Dr Anastasia Christakou (Associate Editor) and Antonia Hamilton (Subject Editor)
openscience@royalsociety.org

Appendix A

Dear Prof Hamilton, dear Dr Christakou,

Please find enclosed a revision of the manuscript (RSOS-180850) “Social cognition, behaviour, and therapy adherence in frontal lobe epilepsy: A study combining neuroeconomic and neuropsychological methods”.

Below I have pasted the original reviewer comments and added our point-by-point responses (marked in boldface).

We would like to thank the reviewers for their excellent comments on the previous draft. This version of the manuscript has benefited greatly from these comments, and we hope that it is now acceptable for publication.

Sincerely,
Andrija Javor – on behalf of all authors

Reviewer: 1

Comments to the Author(s)

This is an interesting paper. Aims are original and clear, so are methods and results and discussion is thorough.

Thank you for your positive overall remarks.

I have some comments and suggestions though.

1. To improve readability I suggest to shorten the methods. For example, sentences like “we subsequently checked whether subjects understood the information...” or “(executive/non executive/no profession...)” can be deleted without subtracting important information and making it easier to follow the paper. An analogous consideration can be done for the long premise in the Result section in the Imaging part (“according to the analysis outlined...”)

We have shortened the method and results sections as suggested in order to improve readability.

2. In the results some data are replicated. I suggest that data referring only to the patients are reported only in the text and data comparing patients and controls are reported only in the table. Whatever the choice, data have to be reported once as duplicates worsen readability.

We have changed the results section as well as table 2 as proposed. Thank you for this comment.

3. In the discussion section I don’t think that the sentence, “ In fact , there is evidence that neurological disorders like Parkinson’s disease can shift the time preference for rewards”, formulated as such, is pertinent.

We have changed this sentence in a way to reflect that the preference for rewards in neurological disorders is not yet understood and evidence is scarce, but that a disease driven change might be possible as reported by the reference publication.

4. The abstract does not mention the imaging part in the design of the study and results. I suppose this is an involuntary omission.

Thank you for highlighting this. We have changed the abstract to make clear that fMRI methodology was used and have included the main result of our imaging analysis (difference in activation of the medial prefrontal cortex between patient and control groups).

Finally I did not understand whether the computer-generated strategy of “player 2” is designed to defect rather than cooperate, thus pushing the subject to change strategy shifting towards defecting over time.

Thank you for pointing out the need to clarify this. In the revised manuscript, we now make clear that the computerized counterpart of the participants was randomly cooperating or defecting, thus was not specifically designed to shift the participants’ strategy.

Reviewer: 2

Comments to the Author(s)

This is a well written, clearly presented study looking at the differences between people with frontal lobe epilepsy and healthy controls on a prisoner’s dilemma task, and the ways in which their responses have an impact on medication compliance. This is an important area of study and the study presents an innovation way of looking at this, with the added potential of clinically relevant findings.

Thank you for your encouraging remarks.

It has a number of limitations, common to all studies of clinical populations, not least the difficulties in accurately measuring compliance with treatment. These are acknowledged by the authors. However there are two larger methodological shortcomings that significantly limit the conclusions that can be drawn from the data.

Firstly, no epilepsy control group was used. It is therefore unclear whether the study findings represent FLE or the impact of epilepsy in general or indeed antiepileptic medications. This is particularly important given the counterintuitive results which indicate that people with FLE cooperate more than controls – could this be due to the social impact of living with a challenging, stigmatising condition? Is this behaviour more common in all people with epilepsy?

We have added a paragraph on this limitation at the end of the discussion section of the manuscript.

Secondly post operative patients appear to have been included in the FLE group, These patients will obviously have very different fMRI activations than non-surgical samples and there is increasing evidence in the fMRI literature that surgery changes patterns of activation. This adds yet another important confounding variable in this already small sample.

Thank you for giving us the opportunity to clarify. In fact, past brain surgery was an exclusion criterion for participation in our study, as we wanted to prevent this being a confounder. We have added a sentence in the method section after our exclusion criteria to clarify this more prominently in the manuscript. Further, we now describe the included patients more precisely in the modified first paragraph of the results section in order to avoid misunderstandings.

Additional changes:

According to journal guidelines, we have moved the ethics statement from the methods section to a separate section after the conclusions.

The competing interests section has also been moved according to journal guidelines.

Appendix B

Dear editors,

The authors would like to thank the reviewers and editors for their time and valuable comments.

Please find enclosed a second revision of our manuscript "Social cognition, behaviour and therapy adherence in frontal lobe epilepsy: A study combining neuroeconomic and neuropsychological methods". We have submitted one version with tracked changes (relative to the last version/first revision) and one without.

Below you will find our point-by-point responses (in bold) to the reviewer's comments. As you will see, we made every effort to address them and therefore hope that the manuscript will now be acceptable for publication.

Best wishes,

Andrija Javor – on behalf of all authors

Reviewer: 1

Comments to the Author(s)
My remarks were adequately addressed.

RESPONSE: Thank you very much for your support in improving our manuscript.

Reviewer: 3

Comments to the Author(s)
Many thanks for the opportunity to review this paper. The authors analysed a sample of 15 patients with frontal lobe epilepsy and 15 healthy controls using neuropsychometry, social cognition testing and an fMRI dilemma game task. Patients exhibited altered activation of the prefrontal cortex, which related to performance in social cognition tests as well as compliance with anti-epileptic drug therapy.

The paper is novel, very interesting, and significantly adds to the literature of social cognition in epilepsy. It will definitely be of interest to the readership of Royal Society Open Science.

REPNSE: Thank you for your positive remarks and your valuable comments to improve the quality of the paper.

I have a number of observations as follows — though most are minor, please note that there are some major concerns regarding the statistical analysis of imaging data, which should be addressed thoroughly.

Abstract

Could the authors specify the number of subjects analysed in the abstract already?

RESPONSE: We have integrated the sample size within the abstract.

Methods

1. "Subjects" section — could the authors be more specific as to what was the cut-off for 'frontal lobe lesions affecting large cortical areas' to grant exclusion of a participant?

RESPONSE: We have reworded this sentence to improve clarity. The new sentence is: "frontal cortical lesions larger than 1 cm in diameter".

2. "Paradigm" section, second line": please change "insure" with "ensure".

RESPONSE: Thank you very much for pointing out this spelling error. This has been adapted.

3. Regarding the cognitive abilities tested by the Trail Making Test A and B-A, the authors should be more specific, as the latter does not merely assess attention. TMT B-A is a measure of 'mental flexibility', which is an executive function, whereas TMT-A more appropriately relates to psychomotor speed. This denominations, consequently, should be adopted in the subsequent manuscript sections too.

RESPONSE: We have adapted the description of the Trail Making Tests A and B accordingly in both methods and subsequent sections.

Results

1. In line with suggestions of previous reviewer, I would recommend further streamlining of the main text. For instance, in the "Behavioural data, neuropsychological variables and questionnaires" section of the manuscript, all z scores and p values for group comparisons could be moved to Table 2. This would also enhance the interpretability of the results reported in the table.

RESPONSE: We have shifted the mentioned data concerning group comparisons into Table 2 and only left information on clinical variables including treatment adherence questionnaires and correlation analyses in the text as Table 2 focuses on demographic, neuropsychological, and neuroeconomic data. Consequently, there is no duplicate data reporting between text and tables in the revised manuscript. This change has also resulted in a modification of the legend of Table 2.

2. When describing first-level imaging data analysis, the authors should specify the type of analysis done (supposedly event-related fMRI) for the convenience of the reader.

RESPONSE: We have added the type of analysis (event-related fMRI analysis) to the text.

3. It would be helpful to provide a figure from a one-sample showing the main effects of task across study groups, thus highlighting the task-associated whole-brain activation map (i.e., an image related to the findings detailed in Table 3).

REPNSE: We have added a new figure (Figure 3) to illustrate the findings detailed in Table 3. We took this opportunity to adapted Table 3 and the corresponding text in the results section to match the analysis type of the other tables. Subsequently, the numeration of the figures has changed slightly (original Figure 3 is now Figure 4).

4. I have concerns on how imaging results were reported with regards to group comparisons. In a 2x2 factorial SPM design, the 'Main effect' of group contrast is helpful to identify areas of group differences in either direction (FLE > CTR or FLE < CTR), is associated with an F statistic, but does not provide information regarding the directionality of effects. The latter is provided by t-contrasts, which are not reported in the manuscript (there are only F-scores in Table 5). How can the authors conclude that activation of the superior frontal gyrus is higher in patients (and not lower), when this was not formally tested? This is a relevant shortcoming and has to be addressed. Appropriate t-contrasts need be carried out both for whole-brain and ROI-based group comparisons.

REPNSE: Thank you for pointing out the need for clarification here. We agree to the fact that an F statistic does not offer information on directionality and we have changed the wording in the results section accordingly. We have done additional t-test analyses on both conditions (choice and results) on a whole-brain and ROI level as proposed. This has been done with and without including other variables that differ significantly between groups as

covariates. These analyses did not yield any significant results. In our view, this shows that (1) the difference between groups shown in the F-contrast (main effect of groups) cannot be explained by a difference in one or the other condition in our sample, but is based on a difference across both conditions, (2) the difference in activation between the groups is not is not sufficiently strong to pass the set threshold in the t-test analyses, and/or (3) might be a result of insufficient sample size. We have adapted the paper in the methods, results, and discussion sections to reflect this thinking and potential limitation of our results.

5. Linked with the above comment — Figure 3 (group comparisons) should be re-done according to what group comparisons actually show.

REPNSE: Due to the results of the t-test analyses depicted above, we have kept our original Figure 3 (now called Figure 4) in the revised manuscript.

6. Always with regards to group comparisons — can the authors confirm that, in the batch system, the following parameter were chosen for the factorial design: (a) Group - Independence: “Yes”, but (b) for Condition (decision/result) —> Independence should be set to “No” — as this is a case of repeated measures design.

RESPONSE: We are happy to confirm the following parameters of our analysis:

Flexible factorial design with 3 factors:

- 1. subject, independence (yes), variance (equal);**
- 2. group, independence (yes), variance (unequal);**
- 3. condition, independence (no), variance (equal).**

In main effects & interactions: main effects (1,2,3) and interaction (2,3).

7. I find the description of the correlation between imaging and behavioural measures a bit confusing. First, it is unclear to me what BOLD % signal change the authors are referring to — was it for the ‘decision’, for the ‘result’ condition, or for an average of both? Second, the whole phrasing of ‘activation change’ is misleading — either ‘signal change’ or ‘activation/deactivation’ should be used to refer to a change in % BOLD, activation change presumes that there’s a dynamic process taking place during the task, which is not the case here.

REPNSE: Thank you again for giving us the opportunity to clarify this section. We have enhanced the language to make clear to what condition the reported BOLD% signal changes are referring to in each sentence of this paragraph. Further, we have changed the term ‘activation change’ to ‘signal change’ as proposed.

Discussion

1. Depending on the actual directionality of the group contrast as discussed in the above comment, the whole section addressing imaging findings (page 13, second paragraph, continuing on the following page) may need to be reformulated

RESPONSE: We have changed this section to reflect the results of our t-test analyses depicted in our response to reviewer comment 4 concerning our results above.

2. In the last paragraph, it would be helpful to put the findings in a broader context, e.g. by mentioning briefly the current state of the art re: social cognition impairment in other epilepsy syndromes with frontal lobe dysfunction, such as JME/IGE (see Giorgi et al., *Epilepsy Research* 2016; Stewart et al., *Seizure* 2018).

REPOSNE: Thank you for pointing out the need to put our results into context. We have added information on Giorgi et al, 2016 and Stewart et al, 2018 in the last paragraph.

Appendix C

Dear editors,

The authors would like to thank the reviewers and editors again for their time and valuable comments.

Please find enclosed a third revision of our manuscript "Social cognition, behaviour and therapy adherence in frontal lobe epilepsy: A study combining neuroeconomic and neuropsychological methods".

Below you will find our point-by-point responses (in bold) to the reviewer's comments. We are very happy that the manuscript is now acceptable for publication.

Best wishes,

Andrija Javor – on behalf of all authors

Reviewer comments to Author:

Reviewer: 3

Comments to the Author(s)

Many thanks for the opportunity to re-review this paper.

I am generally satisfied as to how the authors addressed my previous comments, and I appreciate the restructuring of the discussion.

RESPONSE: Thank you very much for this positive feedback.

A few minor points:

1. For the newly added Figure 3: it would be appreciated if, in the figure legend, the authors could confirm whether the shown activation maps were thresholded at $p < 0.001$, uncorrected (in line, I presume, with what specified in the "Imaging data" paragraph of the Data Analysis section). For consistency, this piece of information should also be added to the legend of Figure 4.

RESPONSE: Our imaging results were thresholded at $p < 0.001$ FWE corrected, reporting clusters greater than 20 voxels only ($k=20$). We added this information to the legends of Figures 3 and 4 and corrected the methods section accordingly.

2. In the main text and figure legend, "one group t test" should be changed to "one sample t test".

RESPONSE: We have changed the wording as proposed.

3. When discussing the full factorial analysis (results section named: "Integrated comparison across groups and conditions", before region of interest analysis): the authors should specify, after "the main group effect", that this was an F-test, to enhance clarity. Such clarification can easily go in brackets.

RESPONSE: We have added the information on the F-test in brackets as recommended.

4. In their response to my point 4 of the Results section, the authors state that additional t test analyses were conducted as requested, both for whole-brain and at a ROI level, and that the latter showed no significant group differences.

However, in the paragraph named “Region of interest analysis” of the Imaging results section (page 10), the first sentence states “ROI-analysis also showed areas of higher activation in patients than in controls in the right and left superior frontal gyrus”. This appears in contrast with what the authors state in their response – could the authors clarify? It should be amended to reflect the fact that there were no significant group differences, I believe.

RESPONSE: Thank you for pointing out the need for clarification here. Indeed, t-tests did not show significant differences in the ROI analyses. The sentence mentioned in this comment is referring to ROI F-test analysis where there indeed was a significant difference. We have changed the wording in this section to enhance clarity.

5. Discussion, 3 line from the beginning – guess the authors mean “shed light” instead of “shade light” ?

RESPONSE: Thank you for finding this typing error that we have corrected in this revision.